# A Phase 1b Adaptive Androgen Deprivation Therapy Trial in Metastatic Castration Sensitive Prostate Cancer

**DOI:** 10.3390/cancers14215225

**Published:** 2022-10-25

**Authors:** Jingsong Zhang, Jill Gallaher, Jessica J. Cunningham, Jung W. Choi, Filip Ionescu, Monica S. Chatwal, Rohit Jain, Youngchul Kim, Liang Wang, Joel S. Brown, Alexander R. Anderson, Robert A. Gatenby

**Affiliations:** 1Department of Genitourinary Oncology, Moffitt Cancer Center and Research Institute, Tampa, FL 33612, USA; 2Department of Integrated Mathematical Oncology, Moffitt Cancer Center and Research Institute, Tampa, FL 33612, USA; 3Fralin Biomedical Research Institute at Virginia Tech, Roanoke, VA 24016, USA; 4Department of Radiology, Moffitt Cancer Center and Research Institute, Tampa, FL 33612, USA; 5Department of Oncological Science, Moffitt Cancer Center and Research Institute, Tampa, FL 33612, USA; 6Department of Biostatistics and Bioinformatics, Moffitt Cancer Center and Research Institute, Tampa, FL 33612, USA; 7Department of Tumor Biology, Moffitt Cancer Center and Research Institute, Tampa, FL 33612, USA

**Keywords:** adaptive therapy, prostate cancer, androgen deprivation therapy, abiraterone, enzalutamide, apalutamide

## Abstract

**Simple Summary:**

Despite early utilization of new hormonal agents (NHA, i.e., abiraterone, enzalutamide and apalutamide) for combined androgen deprivation therapy (ADT) for metastatic prostate cancer, increasingly more men are dying from this disease. While continued development of new drugs is needed, we propose that improved survival of metastatic prostate cancer can be obtained through evolutionarily informed treatment strategies that adjust patient-specific-dosing to their current and past Prostate Specific Antigen (PSA) levels. Compared to the conventional treat-until-progression paradigm, our previous study in metastatic castration resistant prostate cancer (NCT02415621) showed that an on-and-off abiraterone therapy adapted to an individual’s PSA response dynamics provided better cancer control with less drug usage. Here, we report the feasibility of applying this strategy to newly diagnosed metastatic prostate cancer. On-and-off ADTs with luteinizing hormone releasing hormone (LHRH) analog, an NHA, or in combination were based on individual’s testosterone and PSA levels. The current study represents the foundation for future efforts to validate our adaptive therapy in randomized controlled studies for metastatic prostate cancer.

**Abstract:**

*Background*: We hypothesize that cancer survival can be improved through adapting treatment strategies to cancer evolutionary dynamics and conducted a phase 1b study in metastatic castration sensitive prostate cancer (mCSPC). *Methods*: Men with asymptomatic mCSPC were enrolled and proceeded with a treatment break after achieving > 75% PSA decline with LHRH analog plus an NHA. ADT was restarted at the time of PSA or radiographic progression and held again after achieving >50% PSA decline. This on-off cycling of ADT continued until on treatment imaging progression. *Results*: At data cut off in August 2022, only 2 of the 16 evaluable patients were off study due to imaging progression at 28 months from first dose of LHRH analog for mCSPC. Two additional patients showed PSA progression at 12.4 and 20.5 months and remain on trial. Since none of the 16 patients developed imaging progression at 12 months, the study succeeded in its primary objective of feasibility. The secondary endpoints of median time to PSA progression and median time to radiographic progression have not been reached at a median follow up of 26 months. *Conclusions*: It is feasible to use an individual’s PSA response and testosterone levels to guide intermittent ADT in mCSPC.

## 1. Introduction

An estimated 268,490 new cases of prostate cancer with an estimated 34,500 deaths will occur in the United States in 2022 [1]. In 2015, the estimated number of deaths was 27,540. Thus, more men will die from prostate cancer this year compared to 2015 despite the introduction of five life prolonging treatments for metastatic castration resistant prostate cancer (mCRPC) between 2010 and 2013 (sipuleucel-T [2], cabazitaxel [3], abiraterone [4], enzalutamide [5] and radium 223 [6]). While continued drug development is needed, we propose that improved survival of metastatic prostate cancer can be obtained through better utilization of existing agents using evolutionarily enlightened treatment strategies guided by intra-tumor evolutionary dynamics.

We hypothesize that the current treat-until-progression using maximum tolerated dose (MTD) approach is not ideal for clinical management of prostate cancer. In the pivotal trials that led to FDA approval of abiraterone, enzalutamide and apalutamide for treating mCSPC [7,8,9,10], patients were treated continuously at MTD until radiographic or symptomatic progression. Although this approach is necessary to document treatment safety and efficacy for FDA approval, this dosing strategy is often evolutionarily unwise because it maximally selects for resistance and rarely leads to cure. For a clinical setting in which cure is not achievable, we hypothesized that adaptive therapies based on evolutionary dynamics can prolong cancer control with less treatment-related adverse effects and financial toxicity [11,12,13].

In our prior work, we applied a game theory model to guide on and off abiraterone therapy in men with mCRPC (NCT02415621) [14,15]. At the data cut off in January 2022, this strategy significantly improved (*p* < 0.001) median radiographic progression free survival (rPFS) (30.4 months) and median overall survival (OS) (58.5 months) in the adaptive therapy group compared to the 14.3 months median rPFS and 31.3 months median OS in the standard of care (SOC) group [15]. Furthermore, these superior outcomes were obtained with patients receiving abiraterone (on average) just half of their time on trial resulting in an average cost savings of $70,000 per patient per year [16]. Finally, the mathematical model used to design the trial allowed novel analytic methods in which longitudinal trial data permitted revision of key parameters [17,18,19,20,21]. The updated model was then used to simulate the intra-tumoral evolutionary dynamics that led to the observed outcome in each patient in both cohorts [15].

Here, we applied this updated model to a new adaptive therapy trial in metastatic castration sensitive prostate cancer (mCSPC) (NCT02415621). Unlike the study in mCRPC, on-and-off androgen deprivation therapy (ADT) with luteinizing hormone releasing hormone (LHRH) analog, an NHA (abiraterone, enzalutamide, or apalutamide) or in combination were based on individual’s testosterone and PSA levels. The study has already met its primary endpoint of feasibility. The early clinical outcomes on the 16 evaluable subjects are reported here.

## 2. Materials and Methods

### 2.1. Study Design

This is a single center, single arm, phase 1b study funded and sponsored by the H Lee Moffitt Cancer Center and Research Institute. The protocol was reviewed and approved by Moffitt’s Scientific Review Board (SRC) and ADVARRA Institutional Review Board (IRB). It is registered at clinicaltrials.gov under NCT03511196.

### 2.2. Study Population

Men with histologically confirmed metastatic prostate cancer, adequate organ function and ECOG 0-1 performance status were consented and screened within 3 months of starting the first dose of LHRH analog for metastatic prostate cancer. Prostate cancer patients with liver or brain metastases or required opioids for cancer related pain were excluded from the study. Prior LHRH analog therapy (leuprolide, triptorelin, goserelin, relugolix or degarelix) for non-metastatic prostate cancer was allowed if it had been administered for more than a year prior to study enrollment. Patients were also excluded if they had prior treatments with more than 12 weeks of NHAs such as TAK-700/Orteronel, abiraterone, darolutamide, apalutamide or enzalutamide. Patients were enrolled after achieving >75% PSA decline androgen deprivation with 12–16 weeks of LHRH analog and 8–12 weeks of NHA. Both treatments were stopped after enrollment.

### 2.3. Study Procedure

PSA and testosterone levels were measured every 6 weeks and CT and bone scans were performed every 18 weeks while on study. Treatment was restarted if patients developed PSA or radiographic progression per prostate cancer working group (PCWG) 3 criteria [22]. The selection of ADT to restart was based on the patient’s testosterone level at the time of progression: (1) If the testosterone level (T) rose above 100 ng/dL, only the LHRH analog was restarted; (2) if the T was between 50 and 100 ng/dL, only the NHA was restarted and LHRH analog would be added if <50% PSA declined was achieved after 6 weeks of NHA; (3) If the T was below 50, combined therapy with LHRH analog and an NHA was restarted. All treatments were then stopped after achieving a 50% or more PSA decline. For patients who restarted therapy for radiographic progression, partial response or stable disease needed to be documented on the post treatment scans along with PSA response prior to stopping therapy (Figure 1). Patients were taken off study if they developed radiographic progression while on combined ADT with LHRH analog and NHA.

### 2.4. Study Outcomes Assessments

The primary objective of feasibility was measured by the percentage of enrolled subjects who remained on the study at 12 months from their first dose of LHRH analog for metastatic prostate cancer. The study would be terminated early if 2 or more of the first 6 enrolled and evaluable subjects developed on treatment radiographic progression within a year of study enrollment. A subject was not considered evaluable until he had been enrolled and treated per study protocol for 12 months or longer. The secondary objectives of clinical efficacy were defined by median time to PSA progression and median time to radiographic progression. NCI CTAE version 5.0 was used for toxicity assessment. An adverse event (AE) for this protocol was the appearance of (or worsening of any pre-existing) undesirable sign (s), symptom (s), or medical condition (s) occurring after study enrollment even if the event was not considered to be related to starting or stopping study treatments. Abnormal laboratory values or test results constituted adverse events only if they induced clinical signs or symptoms, were considered clinically significant or required therapy (e.g., any hematologic abnormality that required transfusion). Any event that was life threatening, required inpatient hospitalization, prolonged hospitalization (excluding emergency room visits), resulted in persistent or significant disability or incapacity or resulted in death was considered a serious adverse event (SAE). All SAEs related to study treatment were reported and documented on forms as required by institutional guidelines and forwarded directly to the IRB.

## 3. Results

Sixteen evaluable patients were enrolled between April 2019 and June 2021. In terms of racial distribution, subject 103 was black and the other 15 subjects were white. Although the phase 3 trials on upfront combined ADT primarily focused on de novo metastatic prostate cancer [7,8,9,10], the FDA approvals for abiraterone, enzalutamide and apalutamide apply to both de novo and recurrent mCSPC. As shown in Table 1, 25% of the evaluable patients had de novo mCSPC and 5 of the 16 patients had high-risk mCSPC based on the LATITUDE trial criteria [7]. All these 5 patients (102, 105, 106, 107 and 109) had Gleason sum 8 or above and >3 metastatic bone lesions. Patient 108 had lung-only metastases and is the only patient that had neuroendocrine features noted on the tissue diagnosis.

Table 2 shows each evaluable subject’s clinical features before ADT was restarted for the first time after study enrollment. Other than the high-risk definition used in the LATITUDE trial, patients who had metastatic prostate cancer at diagnosis (i.e., de novo metastatic) may have poorer prognoses compared to patients who had definitive local therapy for prostate cancer and then recurred with metastatic disease. Regardless of their risk features and baseline testosterone levels, all 16 evaluable patients had >90% PSA decline at the end of the induction phase with no more than 12 weeks of combined ADT with NHA and LHRH analog. The median PSA nadir at the end of induction phase was 0.09 ng/mL. The testosterone recoveries were rapid. Within 3 months of ceasing therapy, 10/16 patients had testosterone levels >200 ng/mL. Of note, subjects 103 and 108 were off ADT with no PSA or imaging progression for more than 12 months in their first break despite rapid recovery of their testosterone to >200 ng/dL. Subject 103 had pelvic and retroperitoneum lymph node metastases (Figure 2) and 108 had lung only metastases. Both subjects had a complete response to the induction phase of combined ADT with LHRH analog and an NHA. Subjects 111 and 116 had ADT restarted due to PSA progression before testosterone reached 200.

As shown in Figure 3, only two (102, 107) of the 16 enrolled patients developed on treatment radiographic progression and both patients progressed at 28 months from their first dose of LHRH analog for mCSPC. Cell-free tumor DNA (ctDNA) tests at the time of imaging progression showed AR T878A mutation in subject 102 and AR amplification in subject 107. Four subjects had developed on treatment PSA progression at months 12.4 (subject 111), 15.2 (subject 102), 20.5 (subject 109) and 28 (subject 107) from the first dose of LHRH analog for mCSPC. Per protocol, patients 109 and 111 were on continuous NHA plus LHRH analog until they developed imaging progression per PCWG3 (Figure 3). All four patients (102, 107, 109, 111) had ≥6 months prior LHRH analog therapy while receiving definitive or salvage radiation for prostate cancer. These prior exposures to ADT could have contributed to the early development of resistance when ADT was restarted in the metastatic setting. Eight patients (103, 104, 105, 108, 110, 112, 113, 114) were on ADT (including the initial induction phase when ADT was first started for mCSPC) for <40% of the times. The common features they share are the <0.15 ng/mL (mostly <0.1 ng/mL) PSA nadir at the end of combined ADT induction phase (Table 2). The secondary endpoints of median time to PSA progression and median time to radiographic progression have not been reached at the data cut off in August 2022 with a median follow up of 26 months.

## 4. Discussion

As the cornerstone of treatment for metastatic prostate cancer, continuous ADT with LHRH analog is known to cause long-term side effects of fatigue, hot flashes, weight gain, muscle loss, bone loss, sexual dysfunction, gynecomastia and mood swings [23]. Emerging data also raise the concern for cognitive decline with long-term ADT use [24,25]. These long-term sides effects associated with continuous chemical castration are exacerbated by adding NHA to LHRH analog for combined ADT. The goal of intermittent ADT is twofold: to delay treatment resistance and to reduce long term toxicities (including financial toxicity) of continuous ADT.

Among the published randomized studies on intermittent versus continuous ADT for metastatic prostate cancer [26], the phase III SWOG 9346 trial had the largest sample size and the longest follow up [27]. In this study, men with mCSPC were randomized into continuous or intermittent ADT with LHRH agonist treatment after they achieved <4 ng/mL PSA post 7 months of induction therapy with LHRH analog and bicalutamide. The PSA trigger to restart ADT was PSA >20 ng/mL or baseline PSA if initial PSA < 20 ng/mL or PSA >10 ng/mL plus clinician discretion/symptoms. ADT was held after PSA reduced to <4 ng/mL. The study was statistically inconclusive in showing the noninferiority of intermittent versus continuous ADT in terms of its primary endpoint of overall survival (OS). The median OS was 7 months longer in the continuous versus the intermittent arm [27]. In designing our adaptive therapy protocol, we performed computer simulations demonstrating that a 7-month induction period reduced the sensitive population to near-extinction levels producing outcomes from the intermittent arm that are indistinguishable from continuous therapy [14]. Similar limitations apply to other randomized studies in metastatic prostate cancer that used a 6-month induction period and fixed PSA values to restart and then to stop ADT [28,29]. Of note, OS was the primary endpoint of the TAP22 study [28] and no statistically significant difference in OS was observed between intermittent and continuous ADT for mCSPC.

Thus, in the design of this trial, we reduced the induction period of combined ADT to a maximum of 12 weeks to establish the PSA dynamics associated with response but to limit initial therapy with the goal of retaining a significant population of treatment sensitive cells. Furthermore, the cycling time in each patient was determined by the changes in PSA for that individual rather than applying an arbitrary cycle length or PSA value to all patients.

The treatment paradigm of mCSPC has now changed with the approval of adding NHA to LHRH analog for continuous combined ADT until progression. To the best of our knowledge, our phase 1b trial is the first to study intermittent ADT with NHA and LHRH analog. It is also the first study to add serum testosterone level to PSA in the decision making of treatment selection. The design of this study is based on our previously published game theory model of the intra-tumoral evolutionary dynamics in stage IV prostate cancer [13,14,15]. This model is based on three competing tumor subpopulations, i.e., the 3 players in the game of dominance: (i) TP, which expresses high levels of CYP17A1 enabling androgen production from serum precursors (Androgen Receptor/AR+, CYP17+ on IHC); (ii) T+, which requires exogenous androgen (AR+, CYP17− on IHC); and (iii) T−, which is androgen-independent and abiraterone-resistant (AR− and CYP17− on IHC). Computer simulations indicate early treatment withdrawal could be the key to maintain prostate cancer’s sensitivity to ADT. In our study, patients could be enrolled after 8 weeks of combined ADT if >75% PSA decline was achieved. When ADT was restarted for PSA progression, ADT could be discontinued again if >50% PSA decline was achieved. Unlike previous intermittent ADT studies, the decisions on stopping and then reinitiating treatment were based not on a fixed PSA value but on each patient’s PSA and testosterone levels before ADT was started or restarted.

Although limited by the small sample size, homogenous racial distribution and lack of long term follow up, clinical data presented here demonstrate the feasibility of our adaptive therapy approach in both de novo and recurrent mCSPC. Patients (subject 103 and 108) who achieved imaging complete response per PCWG3 after the induction phase seemed to benefit the most from the current adaptive therapy approach in mCSPC. Our game theory model simplified the heterogeneous prostate cancer cells into three competing phenotypes and assumed that each TP or T+ cell produced the same amount of PSA. PSA is one of the downstream targets of the AR signaling pathway and PSA production depends not only on AR status but also on environmental conditions, such as testosterone levels. To improve the accuracy of tumor burden assessment, we are expanding this model to incorporate clinical PSA, testosterone, imaging data, histology data on AR and Cyp17 immunohistochemistry and genomic data on ctDNA, AR amplification and mutations in ctDNA. New models to incorporate spatial heterogeneity are also being used to analyze the trial data [30,31]. Due to the small sample size and lack of long term follow up, we have not detected statistically significant associations between duration of adaptive therapy with Gleason score, choice of NHA (cyp17 inhibitor vs. AR antagonist), mCSPC status (de novo vs. recurrent, high-risk vs. non high-risk). These are important stratification factors for future randomized studies to compare adaptive therapy versus continuous combined ADT in mCSPC.

## 5. Conclusions

Our phase 1b study suggests that it is feasible to use individual’s testosterone level and cancer cells’ PSA responses to guide on and off ADT for mCSPC. Unlike prior studies using cycling ADT, we significantly shortened the induction period to retain a significant population of treatment-sensitive cells to control proliferation of the resistant population. The current trial represents a preliminary investigation of evolution-based therapy using two agents (NHA, LHRH analog) with different but overlapping mechanisms of action, as well as the addition of a second biomarker (testosterone) to the treatment algorithm. The secondary endpoints on median time to PSA progression and median time to radiographic progression have not been met at the time of data cutoff with median time of follow up of 26 months. Clinical and biomarker data from this phase Ib trial along with results from our prior trial in abiraterone therapy in mCRPC are being utilized to improve our mathematical modeling approach for the design of future adaptive therapy trials in metastatic prostate cancer.

## Figures and Tables

**Figure 1 cancers-14-05225-f001:**
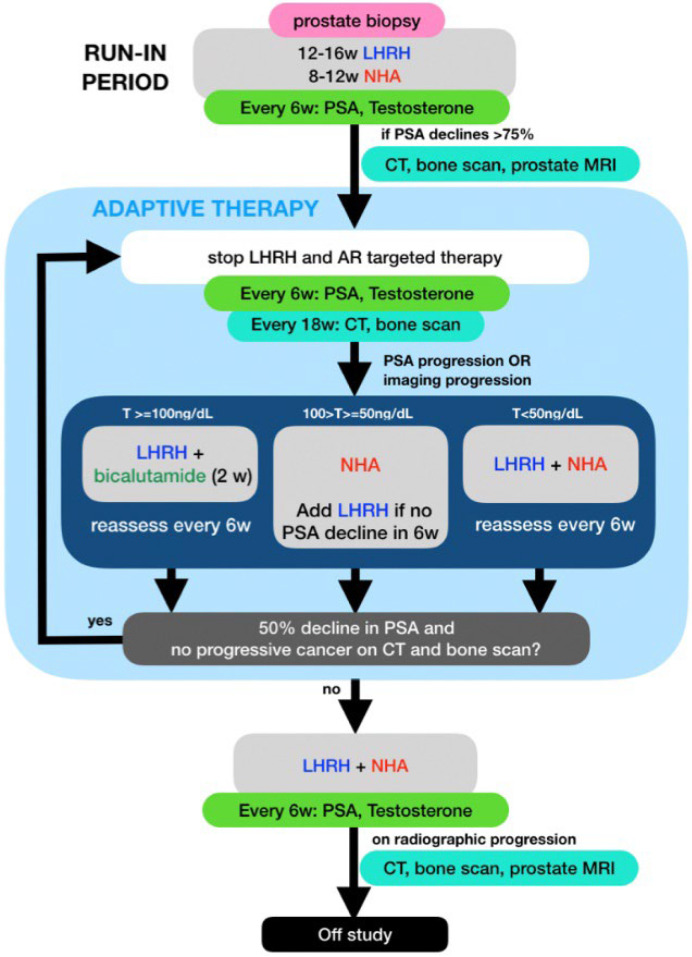
Schema for this mCSPC adaptive therapy trial showing selection of Luteinizing hormone-releasing hormone analog (LHRH) and or a new hormonal agents (NHA) based on PSA, total testosterone (T) and imaging.

**Figure 2 cancers-14-05225-f002:**
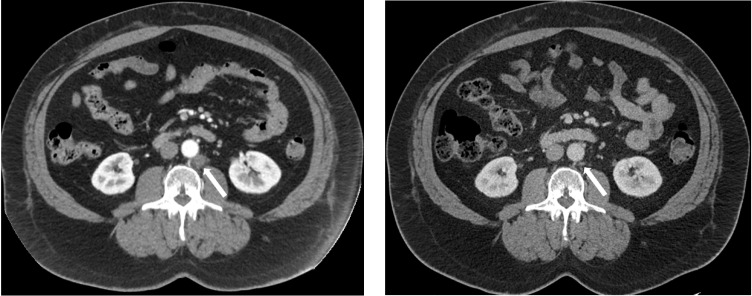
CT images of a retroperitoneal lymph node (white arrow) that has reduced from 1.53 × 1.62 cm (**left**) to 0.40 × 0.60 cm (**right**) after induction ADT in subject 103.

**Figure 3 cancers-14-05225-f003:**
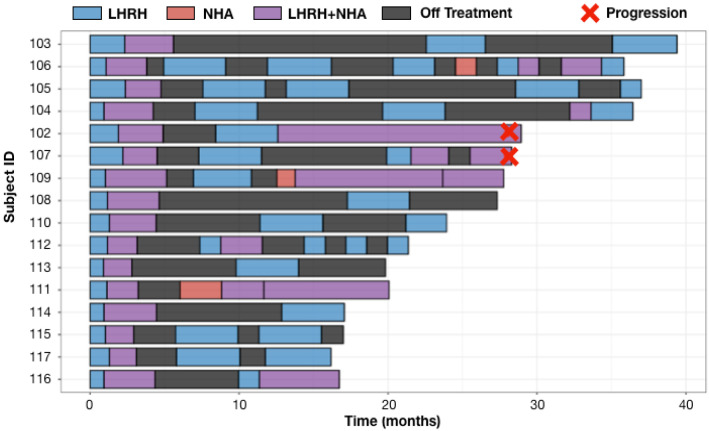
Swimmer Plots for 16 evaluable patients. Safety analysis included one additional patient (101) who withdrew from study treatment after refusing to restart LHRH analog at the time of his first off ADT PSA progression. Adaptive therapy was well tolerated with low frequency of all grades adverse events (AEs) regardless of the attribution. The most common Grade 1–2 treatment-related AEs were arthralgia, hot flashes, back pain and fatigue. One incidence of grade 3 hypertension was reported, which was attributed to abiraterone. All AEs were improved to baseline or resolved with recovery of testosterone while on the treatment break. There were no grade 4 AEs (Table 3).

**Table 1 cancers-14-05225-t001:** Patient characteristics.

	*n* = 16
Age/median [range]	70 (60–78)
PSA ^1^/median [range]	8.5 ng/mL (2.34–46.03 ng/mL)
Gleason sum/median [range]	7 (3 + 4–4 + 50
Prior Prostatectomy/number (%)	5 (31%)
Prior radiation to prostate or prostatic bed	8 (50%)
LHRH analog given with radiation therapy	5 (31%)
De novo metastatic	4 (25%)
Bone only metastases	6 (37.5%)
Visceral/lung only metastases	1 (6%)
Lymph node only metastases	3 (19%)
Bone + lymph node metastases	6 (37.5%)
High risk mCSPC ^2^	5 (31%)

^1^ PSA value prior to first dose of LHRH analog for metastatic prostate cancer. ^2^ High risk mCSPC definition in the phase 3 LATITUDE trial [7] was used, i.e., presence of 2 of the 3 following risk factors: Gleason sum 8 or above, visceral metastases, or more than 3 bone metastases.

**Table 2 cancers-14-05225-t002:** Breakdown of each patient’s risk factors, response to induction ADT, testosterone (T) recovery time and the length of first treatment break/time to restart ADT. ISUP, International Society of Urological Pathology; Abi, abiraterone, Enza, enzalutamide; Apa, apalutamide; NA, not applicable.

	ISUP Group	Risk	De Novo Metastatic	Choice of NHA	Baseline PSA (ng/mL)	Baseline T (ng/dL)	PSA Nadir (ng/mL)	Time to T > 200 ng/dL (months)	Time to Restart ADT (months)
102	4	High	N	Abi	18.51	1078	0.20	<2	5.10
103	2	Low	N	Abi	7.54	256.8	<0.02	<2	17.50
104	2	Low	N	Abi	9.86	251	<0.02	<4	4.10
105	5	High	Y	Abi	30.7	302.4	0.14	<2	4.10
106	4	High	Y	Abi	27.42	321.7	0.08	<1	2.70
107	4	High	N	Enza	2.34	716.4	<0.02	<1	4.10
108	5	Low	Y	Enza	2.99	339.5	<0.02	<2	13.00
109	5	High	N	Enza	8.99	895.9	0.85	<2	2.50
110	2	Low	N	Abi	7.82	478.3	0.03	<4	8.20
111	4	Low	N	Enza	46.03	unknown	1.12	NA	2.70
112	5	Low	N	Enza	2.37	447.8	0.10	<3	4.10
113	3	Low	N	Enza	8.79	678.8	<0.1	<5	6.70
114	3	Low	N	Enza	2.82	230.9	<0.03	4	9.70
115	2	Low	Y	Enza	12.22	876.7	0.17	<1	2.70
116	3	Low	N	Abi	8	475.16	0.55	NA	6.90
117	3	Low	N	Apa	15.43	>1000	0.23	<2	2.90

**Table 3 cancers-14-05225-t003:** Summary of grade 3 or 4 AEs on 17 enrolled patients.

AEs	Grade 3	Grade 4	LHRH Analog	NHA	Other
Hypertension	1 (6%)	0	Unrelated	Related	
Compression Fracture	1 (6%)	0	Related	Unrelated	
Hematuria	1(6%)	0	Unrelated	Unrelated	Radiation Cystitis
Hyperglycemia	1 (6%)	0	Unrelated	Unrelated	Diabetes
Pancreatitis	1(6%)	0	Unrelated	Unrelated	h/o pancreatitis
Sinus Tachycardia	1 (6%)	0	Unrelated	Unrelated	h/o PSVT *
Syncope	1 (6%)	0	Unrelated	Unrelated	Vasovagal

* PSVT, paroxysmal supraventricular tachycardia.

## Data Availability

The data can be shared up on request.

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
