# Peer review of "A Phase 1b Adaptive Androgen Deprivation Therapy Trial in Metastatic Castration Sensitive Prostate Cancer"

_cancers, 2022, doi:10.3390/cancers14215225_

Round 1

Reviewer 1 Report

Congratulations for the authors for their research question and preliminary results. It is a very interesting topic, however some concerns should be addressed;

-In the study objective the number of the previous trial in castration resistant disease is referenced and it is a bit confusing.

- Why 16 patients, but in the design 17 patients were estimated? There is no explanation in the paper for this different numbers.

- It would be better to show Gleason distribution as ordinal with prognostic group classification (ISUP 1-5)

- This sentences “Prostatic adenocarcinoma 150 with 70% acinar and 30% ductal types along with neuroendocrine differentiation were 151 noted on the transurethral resection of the prostate.” should be contextualize. Is it from a specific patient?

- In the paper it is not well described which specific NHA was used. In the trial design is seems to be abiraterone however despite evidence from stampede trial in all mtx, the drug was approved for high-risk patient based on latitude trial, and most evidence is for the novo mtx patients. Could you further explain and discuss this fact? It is important to exactly know the drug used as the mechanism could vary.

- Why did the authors decide PCWG3 criteria, which was developed for CRPC?

Author Response

-In the study objective the number of the previous trial in castration resistant disease is referenced and it is a bit confusing.

Reference 14 is the initial report and reference 15 the updated analysis of NCT02415621. Paragraph 3 is revised to avoid confusion.

- Why 16 patients, but in the design 17 patients were estimated? There is no explanation in the paper for this different numbers.

The sample size for this pilot study is 16 evaluable subjects. Total of 17 subject were enrolled. Subject 101 was deemed not evaluable given he refused to restart androgen deprivation therapy at the time of cancer progression. Subject 101 was included in the safety analysis.

- It would be better to show Gleason distribution as ordinal with prognostic group classification (ISUP 1-5)

Gleason score in table 2 is replaced with ISUP 1-5.

- This sentences “Prostatic adenocarcinoma 150 with 70% acinar and 30% ductal types along with neuroendocrine differentiation were 151 noted on the transurethral resection of the prostate.” should be contextualize. Is it from a specific patient?

Yes, this is referring to subject 108. This sentence is revised to avoid confusion.

- In the paper it is not well described which specific NHA was used. In the trial design is seems to be abiraterone however despite evidence from stampede trial in all mtx, the drug was approved for high-risk patient based on latitude trial, and most evidence is for the novo mtx patients. Could you further explain and discuss this fact? It is important to exactly know the drug used as the mechanism could vary.

The choice of NHA (abiraterone, enzalutamide, apalutamide) is at the discretion of treating physician and patient’s insurance coverage. The specific NHA used in each subject was added to table 2. The choice of NHA did not affect the clinical outcome in this pilot study. As a pilot study on the feasibility of adaptive ADT in mCSPC, we have included both de novo and recurrent metastatic prostate cancer. Although the sample size is small, prior therapy with ≥6 months of ADT is associated with early development of treatment resistance (subject 102, 107, 109 and 111). The above comments on the choice of NHA and de novo metastatic ca were added to the discussion part of the manuscript.  

- Why did the authors decide PCWG3 criteria, which was developed for CRPC?

Although PCWG criteria was developed for CRPC, it was used to define PSA progression and radiographic progression in the Latitude, Enzamet, and Titan trials for metastatic castration sensitive prostate cancer (mCSPC).  We used the PCWG criteria to be consistent with prior published study in mCSPC.

Reviewer 2 Report

Thank you for the opportunity to review the manuscript entitled, " A phase 1b adaptive androgen deprivation therapy trial in metastatic castration sensitive prostate cancer." The clinical topic is important. However, I have several comments to improve the quality of the manuscript.

1.    Could the authors define/make it clearer in the abstract that off study means that patients were removed from the study if they developed progression seen on radiologic imaging?

2.    My biggest concern is the sample size. Is it possible to do a small sensitivity analysis looking at patients with 3+4 Gleason score and compare their averaged outcomes with high-risk patients? Moreover, can the authors build on additional limitations to the small sample size? These limitations include lack of generalizability, possibly homogeneous racial distribution?

3.    It would benefit the paper if the authors provided information on race/ethnicity.

4.    I would suggest tempering the language in the conclusion and begin the sentence that ‘this study suggests’ – primarily due to sample size limitations.

5.    It may be redundant to mention the estimated number of new cases in 2015 – it seems information about mortality is enough?

6.    Could the authors provide units for PSA?

7.    Could the authors provide arrows for figure 2?

Author Response

  1. Could the authors define/make it clearer in the abstract that off study means that patients were removed from the study if they developed progression seen on radiologic imaging?

This is clarified as suggested.

  1. My biggest concern is the sample size. Is it possible to do a small sensitivity analysis looking at patients with 3+4 Gleason score and compare their averaged outcomes with high-risk patients? Moreover, can the authors build on additional limitations to the small sample size? These limitations include lack of generalizability, possibly homogeneous racial distribution.

Thanks for the suggestions. At this time, the follow up is not long enough to correlate Gleason score with time to PSA or radiographic progression (secondary endpoints). We did two sample t test to test for the difference in the length of NHA use versus the total length of follow up. Patients with Gleason 3+4 score did have statistically significant less usage of NHA compared to rest of the patients (p=0.025). We plan to validate this finding after longer follow up with multi-variate analysis.

  1. It would benefit the paper if the authors provided information on race/ethnicity.

Only one black patient was enrolled. Rest of the patients were white. We did not collect data on ethnicity.

  1. I would suggest tempering the language in the conclusion and begin the sentence that ‘this study suggests’ – primarily due to sample size limitations.

The conclusion was revised as suggested.

  1. It may be redundant to mention the estimated number of new cases in 2015 – it seems information about mortality is enough?

This is removed as recommended.

  1. Could the authors provide units for PSA?

Units for PSA were added.

  1. Could the authors provide arrows for figure 2?

This is added.

Round 2

Reviewer 1 Report

all the comments have been addressed

Reviewer 2 Report

The authors have done a nice job responding to my comments. The revised paper is much easier for me to follow. I do not have additional comments.